# The Use of Artificial Hypoxia in Endurance Training in Patients after Myocardial Infarction

**DOI:** 10.3390/ijerph18041633

**Published:** 2021-02-09

**Authors:** Agata Nowak-Lis, Tomasz Gabryś, Zbigniew Nowak, Paweł Jastrzębski, Urszula Szmatlan-Gabryś, Anna Konarska, Dominika Grzybowska-Ganszczyk, Anna Pilis

**Affiliations:** 1Department of Physiotherapy, Jerzy Kukuczka’s Academy of Physical Education, 40-065 Katowice, Poland; z.nowak@awf.katowice.pl (Z.N.); anko@int.pl (A.K.); d.grzybowska-ganszczyk@awf.katowice.pl (D.G.-G.); 2Sport Centrum Faculty of Pedagogy, University of West Bohemia, 301 00 Pilsen, Czech Republic; tomaszek1960@o2.pl; 3Upper Silesian Medical Center, Cardiology Ward, 43-400 Cieszyn, Poland; paweljastrzebski@mp.pl; 4Department of Anatomy, Faculty of Rehabilitation University of Physical Education, 31-571 Krakow, Poland; ulagabrys1957@tlen.pl; 5Faculty of Health Science, Jan Dlugosz University, 42-200 Czestochowa, Poland; a.pilis@ajd.czest.pl

**Keywords:** cardiac rehabilitation, endurance activity, normobaric hypoxia, exercise tolerance, left ventricle

## Abstract

The presence of a well-developed collateral circulation in the area of the artery responsible for the infarction improves the prognosis of patients and leads to a smaller area of infarction. One of the factors influencing the formation of collateral circulation is hypoxia, which induces angiogenesis and arteriogenesis, which in turn cause the formation of new vessels. The aim of this study was to assess the effect of endurance training conducted under normobaric hypoxia in patients after myocardial infarction at the level of exercise tolerance and hemodynamic parameters of the left ventricle. Thirty-five patients aged 43–74 (60.48 ± 4.36) years who underwent angioplasty with stent implantation were examined. The program included 21 training units lasting about 90 min. A statistically significant improvement in exercise tolerance assessed with the cardiopulmonary exercise test (CPET) was observed: test duration (*p* < 0.001), distance covered (*p* < 0.001), HRmax (*p* = 0.039), maximal systolic blood pressure (SBPmax) (*p* = 0.044), peak minute ventilation (VE) (*p* = 0.004) and breathing frequency (BF) (*p* = 0.044). Favorable changes in left ventricular hemodynamic parameters were found for left ventricular end-diastolic dimension LVEDD (*p* = 0.002), left ventricular end-systolic dimension LVESD (*p* = 0.015), left ventricular ejection fraction (LVEF) (*p* = 0.021), lateral e’ (*p* < 0.001), septal e’ (*p* = 0.001), and E/A (*p* = 0.047). Endurance training conducted in hypoxic conditions has a positive effect on exercise tolerance and the hemodynamic indicators of the left ventricle.

## 1. Introduction

Chronic total occlusion (CTO) is reported in 1 in 4 patients undergoing coronary angiography. It has a negative impact on the long-term prognosis of patients and the function of the left ventricle. Over time, some patients with CTO develop collateral circulation to varying degrees, the purpose of which is to supply blood to the peripheral parts of the heart muscle. It has been shown that the presence of a well-developed collateral circulation in the area of the artery responsible for the infarction improves the prognosis of patients and results in a smaller area of infarction [1,2].

For this reason, it is important to search for pharmacological and non-pharmacological methods that can create collateral circulation in patients with chronic occlusion, thus improving the function of the left ventricle, reducing angina symptoms, and improving exercise tolerance, quality of life and long-term prognosis. Numerous scientific reports show that one of the factors that influences the formation of collateral circulation is physical exercise, and its impact depends on an individually selected dose of exercise, based on, among other things, performance studies [3]. The importance of exercises that are properly planned and carried out, during the second and third stage of training improvement, has been repeatedly confirmed in various scientific reports. However, these exercises were always carried out under normoxic conditions with normal oxygen content (20.94% O_2_) [4,5,6]. Currently, the enormous progress in the field of interventional cardiology drives the search for new and even more effective forms of training that have a more dynamic impact on the development of collateral circulation.

One of the factors influencing the formation of collateral circulation is hypoxia, which induces angiogenesis and arteriogenesis, which in turn result in the formation of new vessels [7,8]. The effectiveness of training under normobaric hypoxia has also been confirmed many times in clinical trials of overweight patients [9,10], geriatric patients [11], and sedentary patients [12]. Therefore, the possibility of using artificial hypoxia in the post-hospital (early and late) cardiac rehabilitation program should be considered, especially using physical exercise performed in conditions of reduced oxygen content, both in normobaric and hypobaric conditions, as an additional method of inducing the development of collateral circulation. The aim of the research was to evaluate the effects of endurance training carried out under normobaric hypoxia in patients with ischemic disease or after a heart attack. This training is most often recommended and used in cardiac rehabilitation.

The following research questions were posed:Does the applied endurance training in conditions of artificial hypoxia improve exercise tolerance, hemodynamic parameters of the left ventricle, and the blood counts most often assessed in normoxic conditions in patients with coronary artery disease or after myocardial infarction?On the basis of the obtained research results, can it be concluded that the use of artificial hypoxia in cardiac rehabilitation can be safe for patients and bring beneficial effects?

Research hypothesis: The use of artificial hypoxia during endurance training improves exercise tolerance and hemodynamic parameters of the left ventricle, as well as some blood morphotic parameters.

## 2. Material and Methods

### 2.1. Participants

Enrolment for the study took place in cardiac rehabilitation facilities in the region of Upper Silesia, where the second (ambulatory) stage of cardiac rehabilitation of patients with coronary disease after acute coronary syndrome treated with angioplasty combined with coronary stent implantation was provided. Maintaining the timing of the experiment required systematic training for 4 weeks. The technical conditions of the hypoxic cabin and the behavior of the experimental conditions in this period allowed for the simultaneous participation of a maximum of 35 people. Thirty-five male patients with diagnosed and clinically documented coronary disease aged 43–74 (60.48 ± 4.36 years of age) qualified for the experiment. Descriptive characteristics of the subjects are presented in Table 1, Table 2 and Table 3.

As established in the study protocol, the pharmacological treatment of patients who qualified for the study was optimized and in accordance with the guidelines for coronary disease management.

Due to high risk of the occurrence of side effects caused by normobaric hypoxia, only patients with stable coronary disease were included. The training program began approximately 2 months after discharge from the hospital.

*Inclusion Criteria*:−patients after acute coronary syndrome and angioplasty with stent implantation,−patients with stable coronary disease,−age 35–75,−patients who underwent model A cardiac rehabilitation at least 3 months after the occurrence of acute coronary syndrome,−patients who gave their consent to take part in the study.

*Exclusion Criteria*:−unstable coronary disease,−chronic heart failure during periods of exacerbation,−resistant hypertension,−abnormal exercise test results,−peripheral arterial occlusive disease,−venous thromboembolism,−COPD,−anemia,−disorders of the locomotor system preventing the patient from performing the exercise test,−lack of consent to take part in the study.

### 2.2. Experimental Procedure

Research took place in a laboratory for monitoring physical effort and in an artificial hypoxia chamber (AIR4US, Wichary Technologies, Pyrzowice, Poland) in which four stations for bicycle training were set up. During the experiment, apart from the patients and two training supervisors, in order to ensure safety, there was also a paramedic with full resuscitation protection. The rehabilitation program included endurance training in the conditions of normobaric hypoxia corresponding to an altitude of 2000 m above sea level (O_2_ level = 16.8%, temperature = 19.9 °C, CO_2_ level = 1514 ppm, humidity = 33.2%, atmospheric pressure = 993 hPa). Each patient’s stay in the cabin lasted about 70 min (30 min of acclimatization—sitting on a chair and 40 min cycling), 5 times a week for 21 days.

The training regimen comprised 5 min of warm-up, 30 min of basic training (interval), and 5 min of cooling down—a total of 40 min. The selection of training loads was made on the basis of an initial exercise test conducted before the start of the experiment. Patients who obtained the result >7 MET, which proves good exercise tolerance, without changes in the ECG record and without cardiovascular complaints (NYHA I), were included in the training program.

Progressive high-intensity interval training (HIIT) with moderate cycle duration (1–3 min) and intensity of 80–90% HR peak was used, in line with the recommendations of Gayd et al. [13] and the observations of Burtscher [14]. After 22 training units, all preliminary tests were repeated. 

The following procedure was carried out before commencing the training program and immediately after its completion: electrocardiographic submaximal exercise test on a treadmill (six-stage Bruce protocol: stage 1 =2.7 km/h, 10%, stage 2 = 4.0 km/h, 12%, stage 3 = 5.5 km/h, 14%, stage 4 = 6.8 km/h, 16%, stage 5 = 8.0 km/h, 18%, stage 6 = 8.8 km/h, 20%). The following physiological variables were measured: test duration (min), distance covered (m), energy cost (MET), heart rate at rest (HR_rest_; 1/min) and maximum (HR_max_; 1/min), blood pressure at rest and maximum—systolic (BPS_rest_, BPS_max_; mmHg), diastolic (BPD_rest_, BPD_max_; mmHg), peak minute ventilation (VE; 1/min), breathing frequency (BF; 1/min), peak oxygen consumption (VO_2peak_) per kilogram of bodyweight. These variables were repeatedly used in studies of the level of exercise tolerance and the effectiveness of rehabilitation programs [4,15,16,17]. Spiroergometric parameters were determined with the CORTEX portable METAMAX 3B gas analyzer exercise test using the Excalibur Sport cycle ergometer (Lode, Groningen, The Netherlands).

Prior to use, the system was turned on for at least 20 min, and then calibrated prior to every test according to the manufacturer’s recommendations. This involves first calibrating the gas analyzers using a reference gas (14.97% O_2_, 4.96% CO_2_, balance N_2_: ±0.02% absolute, Hong Kong Specialty Gases), and then verifying the calibration against ambient air. Secondly, volume calibration was performed using a standardized 3-L syringe (5530 series, Hans Rudolph, Inc., Shawnee, KS, USA). To avoid potential gas leakages known to be problematic with facemasks, all participants wore a nose clip and had a mouthpiece attached to the MM3B turbine.

*Blood analysis*: In order to perform peripheral blood counts and electrolytes, about 5 mL of venous blood was collected. The material was collected before the spiroergometric test on the first and last day of the experiment. The following biochemical parameters were analyzed: white blood cells (WBC), red blood cells (RBC), hemoglobin (HGB), hematocrit (HCT), platelets (PLT)—Sysmex K400, cytokines—TNFα, IL β, IL 10 (Tecan, ELISA kit). These variables were also analyzed in other studies involving cardiac patients [6,18,19].

*Lactic acid concentration*: In order to perform the lactic acid test, the nurse drew approximately 200 µL of blood from a fingertip. Lactic acid measurements were taken before the exercise tolerance test, as well as 4 min after exercise (Biosen C-line Clinic, EKF-diagnostic GmbH, Barleben, Germany).

*Two-dimensional ultrasound heart test, measured hemodynamic parameters:* left ventricular end-diastolic dimension (LVEDD; mm), left ventricular end-systolic dimension (LVESD; mm), left ventricular end-systolic volume (LVESV; mL) as per the following formula: LVESV = 7/(2.4 + LVESD) × (LVESD), left ventricular end-diastolic volume (LVEDV; mL) as per the following formula: LVEDV = 7/(2.4 + LVEDD) × (LVEDD), left ventricular ejection fraction (LVEF; %). The following tissue Doppler imaging (TDI) indices were also assessed: E wave (m/s), A wave (m/s), lateral e’ (cm/s), septal e’ (cm/s), E/e’, A/e’, mitral annulus peak systolic excursion MAPSE (Cm)-GE Vivid Q. These variables were analyzed in other exercise studies of cardiac patients [20,21].

### 2.3. Data Analysis

In order to perform statistical analysis for the study, OpenOffice 4.0.1 (Appache Software Foundation, Delaware, DE, USA), StatSoft Statistica (StatSoft Poland, Krakov, Poland) and GraphPad Prism 6.07 software (GraphPad Software, San Diego CA, USA) was used. The Shapiro–Wilk test and histograms depicting frequency distribution of the studied variables were used in order to evaluate the compatibility of their empirical distribution. The homogeneity of variance was measured before the analysis using the Brown–Forsyth test. The statistical tools used to test the statistical hypotheses were:Parametric variance analysis with repeated measurements for variables whose distribution is compatible with normal distribution and the variances of the studied groups are homogeneous,Friedman’s non-parametric variance analysis with repeated measurements for variables whose distribution is not compatible with normal distribution or the variances of the studied groups are not homogeneous.

For statistically significant results suggesting that median changes in parameter values at varying altitudes differ, the post-hoc Turkey test for variables of normal distribution and homogeneous variances and the Dunn–Bonferroni test for variables exhibiting non-normal distribution and non-homogeneous variances were used. The adopted level of significance for the verification of statistical hypotheses was α = 0.05.

## 3. Results

### Exercise Tolerance

After the end of the training program, a significant improvement in exercise tolerance was achieved in the form of an increase in the duration of the test and the distance covered, as well as higher maximum values of heart rate, systolic blood pressure, and breathing frequency (Table 4).

A significant increase in lactate levels was found in the post-exercise study after completing the training program (Table 5).

Blood morphology parameters improved, but without statistical significance. A statistically significant change was found only for the TNFα cytokine level (Table 6).

There were statistically significant changes in the following parameters: LVEDD, LVESD, LVEF BP, lateral e’, septal e’, E/A (Table 7).

## 4. Discussion

Rehabilitation of patients after myocardial infarction aims at improving exercise tolerance, which affects quality of life, and reducing the risk of cardiovascular events. The use of hypoxia combined with training is an innovative approach to the problem. So far, only studies on the effect of intermittent stimulation with hypoxia on the improvement of exercise tolerance have been conducted [14,22]. The analysis of the body response of patients after myocardial infarction to hypoxia caused by low partial pressure of oxygen has been undertaken by many researchers [23,24,25,26]. They mainly focused on the assessment of safety related to staying in hypoxic conditions and the range of its tolerance.

In the authors’ own research, the patients were subjected to conditions corresponding to the altitude of 2000 m above sea level, using artificial normobaric hypoxia, in which the atmospheric pressure was constant, with a reduced oxygen content of 16%. In previous studies carried out in a similar group of patients, it was found that staying in conditions of normobaric hypoxia corresponding to this height, even without the use of effort, gives the best physiological, hemodynamic and biochemical response of the left ventricle [21]. Due to the innovative nature of the research and the risk of possible cardiovascular events resulting from the experimental conditions, the study involved patients whose level of exercise tolerance during the initial exercise test was higher than 7 MET. The reason for choosing such a height above sea level was the fact that the majority of people practicing mountain hiking or skiing most often use tourist resorts located at this altitude. Research carried out in the Alps showed that skiing or mountaineering people with a history of heart attack, hypertension or coronary heart disease were more likely to experience sudden cardiac arrest when staying at high altitudes. It has also been found that up to a height of 2500 m above sea level there is no increased risk of sudden cardiac death [27,28].

The information obtained could be used in the future by doctors and physiotherapists who program rehabilitation for their patients and could help in answering whether going to the mountains is safe for patients with coronary heart disease. It may also be an introduction to a more detailed analysis of the impact of specific high mountain conditions on patients with diagnosed heart disease.

No hypoxic cardiovascular events were recorded during the conducted experiment. All patients completed the training with a significant improvement in exercise tolerance and left ventricular hemodynamic parameters.

### 4.1. Electrocardiographic Exercise Test

Spiroergometry was performed using the cardiopulmonary exercise test (CPET) before and after the end of the 22-day training program. The obtained results clearly confirmed a significant improvement in exercise tolerance. In relation to the initial examination, a significant extension of the test duration was demonstrated (9.92 ± 1.88 vs. 11.22 ± 2.29; *p* < 0.001) and, as a consequence, the distance covered (585.62 ± 169.99 vs. 714.66 ± 226.47; *p* < 0.001). Longer working time was also associated with higher maximum heart rate values (134.02 ± 14.01 vs. 138.57 ± 17.29; *p* = 0.039) and systolic blood pressure (168.25 ± 25.84 vs. 182, 11 ± 33.60; *p* = 0.044). The analysis of respiratory rates assessed during the study showed a significant increase in minute ventilation (86.61 ± 18.82 vs. 94.78 ± 18.87; *p* = 0.004) and breathing frequency (34.38 ± 5.53 vs. 38, 42 ± 6.18; *p* = 0.004).

The daily training lasting for about 40 min clearly contributed to the rapid increase in the level of cardiac muscle efficiency. Such an effect is achieved quite rarely under normoxic conditions. It often requires a longer period of improvement. The decisive role is played by the speed of the collateral circulation process following myocardial infarction [2]. Its pace is believed to depend on the type and intensity of the rehabilitation program used. According to some authors, a factor influencing this process is hypoxia. It can have a significant impact on accelerating angio- and arteriogenesis, causing the formation of new vessels [7,8]. The results of the exercise test after the end of training under the conditions of normobaric hypoxia seem to confirm these assumptions and may indicate the need for further research in this direction. Initially, it was assumed that peak oxygen consumption (VO2_peak_) would increase. However, no significant change was found. The results obtained before and after training were not significantly different (27.06 ± 4.21 vs. 27.62 ± 5.12; *p* = 0.360). The reason for this reaction was a slight (ns) post-workout change in the level of hemoglobin as an oxygen carrier.

Before and 4 min after the end of the spiroergometric exercise test, the level of lactate in the blood serum was determined. No significant differences were found in the resting tests, but a significant increase in post-exercise concentration was found (6.46 ± 2.08 vs. 7.84 ± 2.24; *p* < 0.001), which resulted from the prolongation of working time and its increased intensity.

### 4.2. Blood Test

The analyzed parameters of peripheral blood were: the number of white blood cells (WBC), red blood cells (RBC) and platelets (PLT), hemoglobin concentration (HGB) and hematocrit (HCT) values. There were no significant changes resulting from the training carried out. Presumably, it could have been due to too short a time under hypoxic conditions (about 50 min every day). According to Robach et al. [28], a several-week high altitude exposure and physical effort increase erythropoiesis, more red blood cells are produced, and the hemoglobin mass increases. As a result of acclimatization, the increased amount of hemoglobin enables more efficient transport of oxygen to the tissues. This effect is one of the best known adaptive changes in the organism, allowing it to function efficiently under hypoxic conditions. Having more red blood cells makes the blood more viscous, making it harder for the heart to work. This explains why those born at high altitude (e.g., Sherpas) have an enlarged heart muscle that pumps blood more efficiently. Their lungs and tissues have more capillaries, which greatly facilitates the uptake and transport of oxygen [29].

Similarly to our own research, other authors have also reported no differences in the hemoglobin concentration in red blood cells in response to the concentration of oxygen corresponding to an altitude of 3000 m above sea level [30,31], 4000–5000 m above sea level [32], and 4100 m above sea level. Other authors reported an increase in blood hemoglobin concentration, which occurred only on the 14th day (out of 21) of staying under hypoxic conditions [33,34]. An increase in the concentration of hemoglobin, hematocrit and red blood cells was also reported in their other study carried out at an altitude of 1850 m above sea level [34]. Other studies also noted a significant increase in the above-mentioned parameters after the end of the impact of hypoxia [35,36].

According to Klausen et al., training conducted under both hypobaric and normobaric hypoxia can significantly increase the hemoglobin mass only when exposure to hypoxia corresponding to an altitude over 2100 m above sea level lasts more than 14 h a day for at least 3 weeks [37]. The results obtained by the above-mentioned authors were related to a long (several days) stay in high mountain conditions. However, in our research, the duration of stay in conditions of artificial hypoxia was too short for significant changes in blood morphotic parameters to occur. From the biochemical parameters, a significant decrease was observed for interleukin 10 (4.16 ± 0.90 vs. 3.76 ± 0.86; *p* < 0.001). The level of IL-10 in patients after myocardial infarction correlates with the activity of metalloproteinases, which are a group of proteins that digest elements of the extracellular stroma and are a marker of atherosclerotic plaque instability; they contribute to the progression of coronary artery disease and plaque rupture [38,39]. Obtaining a reduction in its level may indicate a beneficial effect resulting from the use of hypoxic conditions during training. This is only a preliminary hypothesis, requiring further research and confirmation of this effect in a wider group of patients, especially with worse initial exercise tolerance and lower ejection fraction. The durability of the observed effect and its impact on further prognosis therefore require clarification.

### 4.3. Echocardiography

Cardiac echocardiography was performed immediately before and after a series of training sessions under hypoxic conditions. To minimize the effect of subjective assessment of echocardiographic parameters, the assessment was performed by one person, and where possible, automatic measurements were used, with emphasis on the quality of imaging. The results clearly show the improvement of the contractility of the left ventricle. Significant changes were observed for LVEDD (49.28 ± 4.96 vs. 45.51 ± 7.51; *p* = 0.002), LVESD (34.31 ± 6.25 vs. 37.28 ± 8.73; *p* = 0.015), LVEF (49.85 ± 6.65 vs. 52.82 ± 8.52; *p* = 0.021), septal e’ (0.07 ± 0.02 vs. 0.08 ± 0.02; *p* = 0.001), lateral e’ (0.08 ± 0.02 vs. 0.09 ± 0.02; *p* < 0.001), and E/A (1.04 ± 0.39 vs. 1.19 ± 0.48; *p* = 0.047).

Due to the lack of scientific reports analyzing the effect of endurance training performed under hypoxic conditions in post-MI patients on the hemodynamic parameters of the left ventricle, the results were compared to the corresponding groups training under normoxic conditions. Despite the small group of patients participating in the study, a significant improvement was achieved in the most important hemodynamic parameters of the left ventricle in terms of training effectiveness. This applies, inter alia, to the left ventricular ejection fraction most often assessed in publications—the main parameter determining the prognosis of a patient after myocardial infarction [40,41,42]. The favorable changes in the lateral e’ and septal e’ indicators confirmed the positive effect of training also in the area of left ventricular diastolic activity. The demonstrated reduction in the lateral dimensions of the left ventricle (LVESD and LVESD) indicates positive remodeling of the heart muscle. The results indicate an improvement in the function of the heart muscle in response to regular physical exercise, and the conditions of hypoxia do not seem to reduce this effect in comparison with classic models of rehabilitation [43].

Based on the conducted research, it can be concluded that endurance training performed in conditions of artificial hypoxia corresponding to an altitude of 2000 m above sea level is safe for patients after a myocardial infarction and brings beneficial effects related to the improvement of exercise tolerance and hemodynamic parameters of the left ventricle. This opens up new opportunities for cardiac patient rehabilitation programs. Due to its innovativeness, it is an introduction to further research in this direction.

The authors are aware of certain limitations of the study. Future research will be undertaken to resolve some quite important issues that were not addressed in the present study. First of all, it is worth comparing the effects obtained after the training cycle under hypoxic conditions with the results carried out in the same way under normoxic conditions, and also determine the duration of the obtained results. It is also worth assessing the course of the saturation curve itself at the moment of entering the cabin, at the peak of exercise and after the end of training throughout the 22-day observation period. The presented material is only an introduction to further observations and analyses.

## 5. Practical Applications

The results of the research showed that performing endurance training in conditions of artificial hypoxia is safe and effective. Highly effective treatment of patients with coronary heart disease or myocardial infarction, combined with training under artificial hypoxia, may be a better alternative to the methods used so far in cardiac rehabilitation. It can also be a form of preparing patients for a possible trip to mountain areas.

## 6. Conclusions

Endurance training conducted in conditions of normobaric hypoxia has a positive effect on the improvement of exercise tolerance and the hemodynamic parameters of the left ventricle.The use of normobaric hypoxia in cardiac rehabilitation is safe and has a positive effect.The obtained results confirmed the assumed hypothesis with regard to exercise tolerance and left ventricular hemodynamic parameters. In the case of blood morphotic parameters, only IL-10 was confirmed.

## Figures and Tables

**Table 1 ijerph-18-01633-t001:** Location of atherosclerotic lesions.

Artery	Number of Patients
LM	4 (11.42%)
RCA	3 (8.57%)
LAD	18 (51.42%)
Cx	3 (8.57%)
D1	1 (2.85%)
OM1	1 (2.85%)
LAD + RCA	3 (8.57%)
OM1 + Cx	2 (5.71%)

LM—left main coronary artery, RCA—right coronary artery, LAD—left anterior descending artery, Cx—circumflex artery, D1—first diagonal, OM1—first obtuse marginal. In more than half of the patients, atherosclerotic changes were related to LAD.

**Table 2 ijerph-18-01633-t002:** Number of stents implanted.

Number	N (%)
0	0 (0%)
1	30 (85%)
2	5 (15%)
3	0 (0%)
4 or more	0 (0%)
Total	35 (100%)

N—number. The greatest number of subjects had one stent implanted.

**Table 3 ijerph-18-01633-t003:** Pharmacological treatment of male participants.

Type of Medication	Number of Patients
β-blockers	27
Clopidogrel	7
Acetylsalicylic acid (ASA)	25
Atorvastatin	6
α-blockers	3
Vitamin K antagonists	7
Angiotensin II receptor blockers (ARB)	8
Metformin	10
Calcium channel antagonists	4
Angiotensin II converting enzyme inhibitors (ACEI)	10
Diuretics	4

**Table 4 ijerph-18-01633-t004:** Spiroergometric test results.

	Before Rehabilitation Cycle	After 24-Day Rehabilitation Cycle	*p-*Value
	X ± SD	X ± SD	
**Duration** [min]	9.92 ± 1.88	11.22 ± 2.29	<0.001
**Distance** [m]	585.62 ± 169.99	714.66 ± 226.47	<0.001
**MET** [mL/kg/min]	7.73 ± 1.19	7.95 ± 1.44	0.214
**HR_rest_** [1/min]	71.71 ± 9.12	70.01 ± 8.81	0.281
**HR_max_** [1/min]	134.02 ± 14.01	138.57 ± 17.29	0.039
**SBP_rest_** [mmHg]	124.14 ± 11.08	124.01 ± 13.10	0.957
**DBP_rest_** [mmHg]	71.00 ± 8.20	70.42 ± 8.34	0.781
**SBP_max_** [mmHg]	168.25 ± 25.84	182.11 ± 33.60	0.044
**DBP_max_** [mmHg]	78.57 ± 10.54	74.35 ± 12.41	0.210
**VE** [1/min]	86.61 ± 18.82	94.78 ± 18.87	0.004
**BF** [1/min]	34.38 ± 5.53	38.42 ± 6.18	0.004
**VO_2peak/kg_** [mL/min/kg]	27.06 ± 4.21	27.62 ± 5.12	0.360

All data are presented as mean values ± standard deviations, *p*—statistically significant level (the lowest assumed level was *p* ≤ 0.05), MET—metabolic equivalent, HR_rest_—heart rate at rest, HR_max_—maximum heart rate, SBP_rest_—systolic blood pressure at rest, SBP_max_—maximum systolic blood pressure, DBP_rest_—diastolic blood pressure at rest, DBP_max_—maximum diastolic blood pressure, VE—minute ventilation, BF—breathing frequency, VO_2_max—maximal oxygen uptake, RER—respiratory equivalency ratio.

**Table 5 ijerph-18-01633-t005:** Blood lactate level.

	Before Rehabilitation Cycle	After 24-Day Rehabilitation Cycle	*p-*Value
	X ± SD	X ± SD	
**Lactates 1** [mmol/L]	1.43 ± 0.53	1.58 ± 0.53	0.155
**Lactates 2** [mmol/L]	6.46 ± 2.08	7.84 ± 2.24	<0.001

Lactates 1—lactates at rest before spiroergometric test, Lactate 2—lactates 4 min after spiroergometric test.

**Table 6 ijerph-18-01633-t006:** Blood morphology.

	Before Rehabilitation Cycle	After 24-Day Rehabilitation Cycle	*p-*Value
	X ± SD	X ± SD	
**WBC**	6.52 ± 1.50	6.20 ± 1.11	0.059
**RBC**	4.91 ± 0.37	4.92 ± 0.34	0.637
**HGB**	14.96 ± 1.03	15.01 ± 0.99	0.563
**HCT**	44.48 ± 2.99	44.25 ± 2.62	0.438
**PLT**	211.22 ± 53.28	214.60 ± 52.21	0.407
**IL 1beta**	34.11 ± 34.66	48.80 ± 69.81	0.060
**IL 10**	4.16 ± 0.90	3.76 ± 0.86	<0.001
**TNF**	14.77 ± 14.11	14.37 ± 17.45	0.666

WBC—white blood cells, RBC—red blood cells, HGB—hemoglobin, HCT—hematocrit, PLT—platelets, IL—interleukin, TNF—tumor necrosis factor.

**Table 7 ijerph-18-01633-t007:** Results of echocardiographic tests carried out before and after the 24-day rehabilitation cycle.

Variable	Before Rehabilitation Cycle	After 24-Day Rehabilitation Cycle	*p-*Value
	X ± SD	X ± SD	
**LVEDD**	49.28±4.96	45.51 ± 7.51	0.002
**LVESD**	34.31 ± 6.25	37.28 ± 8.73	0.015
**LVESV**	51.77 ± 13.70	50.11 ± 12.58	0.460
**LVEDV**	105.62 ± 23.17	108.85 ± 25.13	0.339
**LVEF**	49.85 ± 6.65	52.82 ± 8.52	0.021
**E wave**	0.64 ± 0.17	0.68 ± 0.18	0.132
**A wave**	0.65 ± 0.06	0.63 ± 0.20	0.400
**e’ lateral**	0.08 ± 0.02	0.09 ± 0.02	<0.001
**e’ septal**	0.07 ± 0.02	0.08 ± 0.02	0.001
**E/A**	1.04 ± 0.39	1.19 ± 0.48	0.047
**E/e’**	8.37 ± 2.75	7.64 ± 1.93	0.117
**MAPSE**	14.37 ± 2.88	15.85 ± 4.27	0.050

LVEDD—left ventricular end-diastolic dimension, LVESD—left ventricular end-systolic dimension, LVESV—left ventricular end-systolic volume, LVEDV—left ventricular end-diastolic volume, LVEF—left ventricular ejection fraction, MAPSE—mitral annulus peak systolic excursion.

## Data Availability

The data presented in this study are available on request from the corresponding author

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
