# Peer review of "The Use of Artificial Hypoxia in Endurance Training in Patients after Myocardial Infarction"

_ijerph, 2021, doi:10.3390/ijerph18041633_

Round 1

Reviewer 1 Report

I congratulate authors on their work. Overall, I found the topic timely and clinically important.
My main concerns are:

Introduction:

(a). The introduction needs to be clear what the practical question is that you are trying to address.

(b). How the answer to this question is important to the field as this is not clear or obvious?

(c). How is this study and impactful study and not trivial as this needs more clarity as well. The key issue here is to make sure you set up your approach to the problem.

(d). The approach to the problem is essential in determining and describing the rationale for the study. You have not given a basic rationale for the choices made for the variables used in the study.

Methods and results:

(a). Lack of justification for the sample size.

(b). For scientific reasons, the sample size for trial needs to be planned carefully.

Author Response

Thank you very much for your valuable tips. They are shown in the text in red (blue is the answer to the second reviewer). Answering the question about the number of respondents - Due to the limitations related to the technical possibilities of the artificial hypoxia cabin, but above all the available dates in which it was possible to conduct research, only 35 patients could participate in the experiment. We plan to continue the research in the near future, so the number of participants will certainly increase. We treat the current results as a pilot study.

Literature was also supplemented.

Reply to Reviewer 1’s comments:

I congratulate authors on their work. Overall, I found the topic timely and clinically important.
My main concerns are:

Introduction:

(a). The introduction needs to be clear what the practical question is that you are trying to address.

Response:  Thank you for your advice. Your suggestions were included in the content of the work. Research questions added at the end of the introduction.

(b). How the answer to this question is important to the field as this is not clear or obvious?

 Response:  The required comments were included in the text, as suggested by the reviewer

(c). How is this study and impactful study and not trivial as this needs more clarity as well. The key issue here is to make sure you set up your approach to the problem.

Response:   We hope that the changes we have made to the introduction are satisfactory to the reviewer.

(d). The approach to the problem is essential in determining and describing the rationale for the study. You have not given a basic rationale for the choices made for the variables used in the study.

Response:  Of course, we fully agree with the reviewer's suggestions. the relevant content related to the justification of the selection of the variables has been moved to the Material and Methods chapter

Methods and results:

(a). Lack of justification for the sample size.

Response:  Due to the limitations related to the technical possibilities of the artificial hypoxia cabin, but above all the available dates in which it was possible to conduct the research, only 35 patients took part in the experiment. Research will continue as soon as possible. We treat the current research as pilot studies.

 (b). For scientific reasons, the sample size for trial needs to be planned carefully.

Response: The justification for the suggestion is presented in the comment above

Reviewer 2 Report

Please, see attached document. I have added a table, and it is a reason for no add my comments here. 

Author Response

Thank you for your very insightful assessment and valuable tips. We made a correction as suggested.

The aim of the study was not to analyze the "benefits" but to analyze the "effects" of training, therefore the suggested word was not changed. We didn't know if we would get any benefits after completing the training program.

Replacement of the definition of "endurance training" by "endurance exercise" - the applied program has the character of a training with a specific purpose, duration and content. It was used daily and the individually selected loads were incremental.

Reply to Reviewer 2’s comments:

Dear authors and editor,

Thank you very much for consider me as a reviewer for this study. Overall, it seems a well conducted search, but it lacks from some crucial issues.

I have assessed the risk of bias using modified STROBE survey [1]:

1

2

3

4

5

6

7

8

9

10

Q

1

0

1

1

1

1

1

1

0

1

high

Response: Thank you for you assessment.

However, it is a lack of some specific issues:

Title:

It is only a suggestion, but, it could sound better:

“The benefits of the use of artificial hypoxia in endurance training in patients after myocardial infarction”.

Response: Thank you for valuable tips, however, we think that the use of the word 'benefits' is inappropriate from the point of view of the aim of the work and research methodology. Therefore, we decided that we can change the topic of work by emphasizing the 'effect' of the training. We have not adopted any criteria for the benefit of training effects because it is too extensive a topic.

Following PRISMA guidelines [2], if the “outcomes” are not included, the title would be incomplete.

Moreover, consider change “endurance training” by “endurance exercise”. Maybe, “training” refers to improve performance, where the aim of this treatment are improvements in health.

Response: The program is a training with appropriate content, duration and purpose. The loads were selected individually, based on the results of stress tests. The training program assumed a load progression. An exercise is a single unit, the element of training that does not have the appropriate characteristics required by our project.

Abstract:

It is a lack of background. For the non-expert readers in this topic, the contextualization of the presented objective becomes mandatory.

Response: The required comments were included in the text, as suggested by the reviewer

SBPmax, VE, LVEDD, LVESD, and LVEF are not defined the first time in which they were mentioned.

Response: Thank you for your suggestions. We completely agree with Editor’s opinion. Required comments were included in the text.

Key words:

The key words should not repeat the words that appear in the title or in the abstract. Please, change “myocardial infarction” and “endurance training”.

Response: We agree with the reviewer's suggestion. Appropriate changes were introduced in the text.

Introduction:

The first sentence of the second paragraph is too long. In addition, it lacks of references.

Response: Appropriate changes were introduced in the text.

The second phrase of the second paragraph is too long. And in addition, at the end, it mentions “performance studies”, where it is a unique citation. Please, add references.

Response: Appropriate changes were introduced in the text.

“The importance of properly planned and carried out exercises during the second and third stage of training improvement, including general conditioning, resistance and endurance exercises, has been repeatedly confirmed in various scientific studies.” What are these studies?

Response: Appropriate changes were introduced in the text. Citation added

However, these exercises were always carried out under normoxic conditions. Where are the references?

Response: Citation added in the text.

The second and the third paragraphs should appear together.

Response: Appropriate changes were introduced in the text.

The paragraph “studies conducted on dogs...” sound estrange because it is an study conducted in humans, not in animals, and since it a wide searched topic, studies developed with animals not defined the state of the art. So, please, considered delete this paragraph.

Response: Of course, we agree with the Editor. Indicated text was delated.

I think that it is not clearly conducted the final part of the introduction. The authors said that several studies have assessed the normobaric hypoxia in sport. Hence, where is the novelty of the proposal?

Response: We found that the descriptions related to the research of athletes refer to healthy people with a much higher level of exercise tolerance than the studied patients. We have therefore removed this content.

Please, if this article is a novel proposal due to the application of resistance exercise, consider add a brief explanation about what has been done in sport and hypoxia, and add sentences such as: “..., but, to the authors knowledge, it is no an study that ...”.

Response: In line with the above answer, the text about athletes has been removed because our work has examined patients, not athletes. The reviewer's suggestion is absolutely justified.

Add a hypothesis, and a clear objective (as in the abstract).

Response: Hypothesis were added

The sentence “Selected hemodynamic parameters of the left ventricle and blood counts of patients with coronary heart disease or a history of myocardial infarction were measured.” belongs to the methodology.

Response: We agree with the reviewer's suggestion. Appropriate changes were introduced in the text.

Methodology:

Since I am not an expert on this topic, I do not have the capacity to value it.

Results:

The test which refers to a table should be before the table. So, change over the results this issue.

Response: We agree with the reviewer's suggestion. Appropriate changes were introduced in the text.

Discussion:

It lacks from references. Please, add references in all ideas from other articles. For example: “In the authors' own research, the patients were subjected to conditions correspond-ing to the altitude of 2,000 m above sea level, using artificial normobaric hypoxia, in which the atmospheric pressure was constant, with a reduced oxygen content of 16%.” This sentence support your conclusions, but, where this idea was reported?

Response: Appropriate citation has been added to the discussion body

Conclusions:

Since in this study was not conducted in 2,000 m above sea level(despite the authors say that the conditions are similar), the conclusions are not suitable. The conclusions should be “endurance training conducted in hypoxic conditions has a positive effect on the improvement of exercise tolerance and hemodynamic indicators of the left ventricle.”, as the authors mentioned in the abstract.

Response: Of course, conclusion was completed.

Round 2

Reviewer 2 Report

The article may be accepted in its current form.